# Understanding Antimicrobial Resistance (AMR) Profiles of *Salmonella* Biofilm and Planktonic Bacteria Challenged with Disinfectants Commonly Used During Poultry Processing

**DOI:** 10.3390/foods8070275

**Published:** 2019-07-22

**Authors:** Myrna Cadena, Todd Kelman, Maria L. Marco, Maurice Pitesky

**Affiliations:** 1UC Davis School of Veterinary Medicine, Department of Population Health and Reproduction, Cooperative Extension, One Shields Ave, Davis, CA 95616, USA; 2UC Davis, Department of Food Science and Technology, One Shields Ave, Davis, CA 95616, USA

**Keywords:** *Salmonella*, biofilm, disinfectants, poultry, transcriptome, resistance

## Abstract

Foodborne pathogens such as *Salmonella* that survive cleaning and disinfection during poultry processing are a public health concern because pathogens that survive disinfectants have greater potential to exhibit resistance to antibiotics and disinfectants after their initial disinfectant challenge. While the mechanisms conferring antimicrobial resistance (AMR) after exposure to disinfectants is complex, understanding the effects of disinfectants on *Salmonella* in both their planktonic and biofilm states is becoming increasingly important, as AMR and disinfectant tolerant bacteria are becoming more prevalent in the food chain. This review examines the modes of action of various types of disinfectants commonly used during poultry processing (quaternary ammonium, organic acids, chlorine, alkaline detergents) and the mechanisms that may confer tolerance to disinfectants and cross-protection to antibiotics. The goal of this review article is to characterize the AMR profiles of *Salmonella* in both their planktonic and biofilm state that have been challenged with hexadecylpyridinium chloride (HDP), peracetic acid (PAA), sodium hypochlorite (SHY) and trisodium phosphate (TSP) in order to understand the risk of these disinfectants inducing AMR in surviving bacteria that may enter the food chain.

## 1. Introduction

*Salmonella* is a major foodborne pathogen worldwide and is highly associated with contaminated poultry products. In the United States alone, the Centers for Disease Control and Prevention (CDC) estimates that *Salmonella* causes approximately 1.2 million foodborne illnesses, 23,000 hospitalizations and 450 deaths per year [1]. In addition to causing foodborne illness, *Salmonella* isolates from poultry products and processing plants have been found to be both tolerant to disinfectants and resistant to antibiotics despite not being challenged with antibiotics during poultry production and/or processing [2,3,4,5]. Furthermore, studies have shown positive correlations between tolerance to disinfectants and resistance to antibiotics in poultry products [4,5]. Growing concerns over disinfectants conferring cross-protection to antibiotics has increased focus on understanding the mechanisms of antimicrobial resistance (AMR) in bacteria, and specifically foodborne pathogens [6]. The present review aims to understand how commonly used disinfectants such as hexadecylpyridinium chloride (HDP), peracetic acid (PAA), sodium hypochlorite (SHY) and trisodium phosphate (TSP) may confer AMR in *Salmonella* in order to evaluate the risk of these disinfectants increasing AMR in surviving *Salmonella*.

Biofilms are organized structures of bacterial cells that produce a self-encasing polymer extracellular matrix and can adhere to biotic (living) and abiotic (inert/nonliving) surfaces [7]. As opposed to the more commonly studied and well understood planktonic (free-floating) form, biofilms are the predominant form of bacterial growth. It is estimated that 80% of all infections in humans are thought to be of biofilm origin [8]. Various abiotic substrates, such as Teflon™, stainless steel, rubber and polyurethane can support *Salmonella* biofilm adherence and growth [9,10], which are regulated by various environmental conditions such as pH, temperature and NaCl concentration [11]. Within a poultry processing facility setting, *Salmonella* and *Campylobacter* biofilm formation is facilitated by the presence of meat juice on abiotic surfaces under static and flow conditions [12]. Formation of biofilms provides ecologic advantages to the enclosed bacteria, including protection from the environment (e.g., temperature, pH and osmotic extremes, UV light exposure, desiccation), increased nutrient availability, metabolic enhancement, and facilitation of gene transfer [13]. Additionally, biofilm formation confers increased antimicrobial resistance through a variety of mechanisms. From a practical perspective, biofilm cells are 10- to 1000-fold less susceptible to anti-microbial agents than the planktonic form of the same bacterium [8,14,15]. In poultry processing plants, the use of sub-optimal concentrations of a commonly applied biocide (peracetic acid) has been demonstrated to facilitate the persistence of *Salmonella* biofilms [16]. All of these characteristics appear to be facilitated at least in part by an intercellular communication mechanism known as quorum sensing—small signal molecules called autoinducers exchanged between bacteria as a function of population density. These signal molecules can regulate expression of numerous genes, including those associated with biofilm adherence, metabolism, and virulence. The development of inhibitors of such factors may be key to controlling biofilm formation and pathogenicity [10,13,17,18,19].

## 2. Disinfectants Commonly Used during Poultry Processing

In the most general sense, poultry processing can be divided into two phases—first processing and second processing (see Figure 1) [20]. First processing consists of carcass receiving to chilling. This step includes scalding, defeathering and evisceration. Second processing encompasses carcass chilling to shipping. This step includes packaging and may include carcass/parts processing.

Moreover, during processing, carcasses that are contaminated with fecal matter or digestive tract content as determined by visual inspection right after evisceration may still pass inspection if reprocessed properly [21]. Two methods of reprocessing include on-line reprocessing (OLR) and off-line reprocessing (OFLR). Processing plants may use either or both forms of reprocessing. In OLR, contaminated carcasses can be reprocessed manually by trimming away contaminated parts and/or treated with an antimicrobial along with poultry carcasses that are not contaminated on the main line. In contrast, OFLR entails taking out contaminated carcasses off the main line where they can be reprocessed manually by trimming away contaminated parts and/or treated with an antimicrobial, away from the visually uncontaminated carcasses. Processing plants may use either or both forms of reprocessing. Then, before carcass chilling, carcasses are visually inspected again for fecal contamination using guidelines provided by the Food Safety Inspection and Inspection Service Directive in order to comply with the zero-tolerance standard that requires carcasses to be free of fecal contamination before entering the chiller tank [22,23]. While disinfectants are used in both first and second processing, their application, contact time and temperature may differ. Similarly, disinfectants approved for use during OLR and OFLR may differ in the application, contact time and temperature [24]. For the purposes of this review, only disinfectants commonly used during poultry processing will be reviewed, as intervention protocols may vary greatly across poultry processing plants. Popular disinfectants include hexadecylpyridinium chloride (HDP), peracetic acid (PAA), sodium hypochlorite (SHY) and trisodium phosphate (TSP) [25,26]. While PAA is the most commonly used disinfectant in both pre- and post-chill applications, HDP is most commonly used in post-chill applications when drench cabinets are used [27]. Additionally, both HDP and PAA can be used for OLR and OFLR [24]. SHY and TSP are also commonly used in post-chill applications [25]. In terms of reprocessing, SHY is only allowed in OFLR, although when it is used in combination with other disinfectants it can be used in OLR as well. In contrast, TSP is only allowed in OLR. Figure 1 provides a schematic overview of poultry processing along with information on HDP, PAA, SHY and TSP.

## 3. Poultry Processing Methods Conferring Biocide Tolerance

Currently, studies suggest that biocide tolerance provides cross-protection to various antimicrobials, including antibiotics [28]. Both repeated exposure and sub-inhibitory concentrations of biocides have been shown to allow bacteria to adapt to biocides resulting in biocide-tolerant, antibiotic-resistance bacteria [28,29,30]. As seen in Figure 1, pathogens such as *Salmonella* may undergo repeated exposure to disinfectants during poultry processing especially if they are allowed in OLR and OFLR. In addition, challenging *Salmonella* with sub-inhibitory concentrations of disinfectants at the processing plant can occur due to the presence of high loads of organic material, inadequate distribution, high prevalence of biofilms, inadequate mixing, preparation, and concentration of biocides. Chicken carcasses have high amounts of organic material that can inactivate certain classes of disinfectants, including quaternary ammonium compounds (HDP) and halogens (SHY) [31]. Additionally, *Salmonella* can adhere to areas that are not easily accessible to disinfectants, in places such as crevices and feather follicles on poultry skin [31,32].

With respect to biofilms, the extracellular matrix limits access to stressors, thereby providing some protection against disinfectants and antibiotics that allow *Salmonella* to persist on biotic (e.g., live birds) and abiotic (e.g., stainless steel) surfaces [6]. While biofilms are more prevalent than the easier-to-kill planktonic form in processing plants, the common methodology for evaluating the efficacy of disinfectants requires identifying the inhibitory concentration and or log reduction of the bacteria in their planktonic form [33]. Therefore, if the tests used to determine effective disinfectant concentrations are done on planktonic bacteria, it is possible that the concentrations are sub-inhibitory in the processing plant. Shah et al. demonstrated that when *Salmonella* Typhimurium was preadapted to cold stress, it was tolerant to subsequent acid stress [34]. Therefore, it is possible that *Salmonella* passing the chilling stage may also be harder to kill and require a higher concentration of disinfectant at the post-chill tank. 

Overall, processing methods may be priming bacteria to stressors found throughout the slaughtering process resulting in surviving bacteria that are tolerant to biocides and resistant to antibiotics. Therefore, it is becoming increasingly important to understand how methods of pathogen control in food processing can be improved in terms of reducing tolerance to disinfectants and resistance to antibiotics. The specific mechanisms that can confer biocide tolerance and antibiotic resistance in bacteria exposed HDP, PAA, SHY and TSP will be discussed next. 

## 4. Proposed Mechanisms of Bacterial Resistance to Antimicrobials Induced by Disinfectants Used During Poultry Processing

Cross-resistance, or resistance to a variety of substances via a physiological adaptation, as opposed to genetic linkage as is the case with co-resistance, is an important mechanism of bacterial resistance to antimicrobials [28]. Examples of cross-resistance mechanisms include reduced cell permeability, production of neutralizing enzymes, target alteration and overactive efflux pumps which can pump out a broad-spectrum of substances including antibiotics, biocides and other inhibitors out of the cell and create multidrug resistant (MDR) bacteria [6,35].

In particular, overactive efflux pumps and changes to the outer membrane have been proposed as broad-spectrum mechanisms conferring tolerance and/or resistance to antimicrobials in *Salmonella* after exposure to HDP and TSP [4,36] (Table 1). To evaluate these proposed mechanisms, Mavri et al. exposed TSP- and HDP- adapted *Campylobacter jejuni* and *Campylobacter coli* (Gram-negative bacteria like *Salmonella*, which share a superfamily of efflux pumps known as Resistance-Nodulation-Division [37,38]) to efflux pump inhibitors and evaluated their outer membrane proteins [30]. Results showed that TSP- adapted *Campylobacter jejuni* and *Campylobacter coli* had a weaker adaptive resistance to TSP and weak cross-resistance to antibiotics compared to HDP-adapted *Campylobacter jejuni* and *Campylobacter coli* [30]. The authors proposed that different efflux systems play a role in cross-resistance due to different modes of action of the disinfectants resulting in different levels of cross-resistance [30]. Additionally, TSP caused greater reduction in outer membrane protein (OMP) content than HDP, resulting in the most damaging effects on bacterial cells [30]. It was also noted that some strains of *Campylobacter* displayed increased susceptibility to biocides after repeated exposure—Mavri et al. proposed that some cell envelope modifications may actually promote biocide uptake [30]. Along with linking efflux pumps to cross-resistance, this study revealed that mechanisms involved in biocide (e.g., triclosan, benzalkoniumchloride, hexadecylpyridinium chloride, chlorhexidine diacetate and trisodium phosphate) adaptation are unique for various strains of *Campylobacter* as opposed to it having only one species-specific mechanism [30]. These findings suggest that utilizing a serotype-specific or even a strain-specific approach to select disinfectants is becoming increasingly important. 

In addition to AMR and MDR, efflux pumps have also been associated with increased invasion in *Salmonella* [43]. For example, in addition to regulating resistance to fluroquinolones in *Salmonella*, the acrAB operon—part of the AcrAB-Tolc multidrug efflux pump—has also been shown to be upregulated during sub-inhibitory exposure to the bile salt sodium deoxycholate (DOC), particularly during exponential growth [43,44]. DOC at high concentrations exhibits biocidal-like activity including disruption of cell membranes, denaturation of proteins and oxidative DNA damage [44]. By adapting to DOC, *Salmonella* Typhimurium can then proliferate and continue to invade the host, while strains lacking AcrAB-Tolc were unable to adapt to DOC [44]. From a food safety perspective, cross-resistance imposes a food safety hazard in that repeated exposure to biocides can potentially induce biocide tolerance, AMR and increased virulence in bacteria entering the food chain [28]. 

In contrast to the increased susceptibility of *Campylobacter* to biocides after exposure, SHY has been shown to induce biofilm production in *Pseudomonas aeruginosa* (Table 1), also a Gram-negative bacteria like *Salmonella* [42]. As discussed previously, biocide tolerance and antibiotic resistance can be attributed to biofilms, as the extracellular matrix provides the cells protection against disinfectants and antibiotics [6], while the clustering of cells may facilitate the transfer of antimicrobial resistance genes via horizontal gene transfer [45]. Extracellular DNA may also play a role in the proliferation of biofilms in *Salmonella* and other bacteria: *Staphylococcus epidermidis* biofilm has been shown to have a strong binding affinity to vancomycin thereby limiting access to cells [46,47]. Additionally, RNA-sequencing analysis of planktonic and biofilm *Salmonella*, indicates that gene expression patterns differ between the two forms under the same acid stress [48,49]. Furthermore, RNA-sequencing suggests that in *Salmonella* Typhimurium the same environmental stressors results in upregulation of virulence genes in the planktonic form—priming that population for host invasion rather than for environmental survival as it does for the biofilm counterpart [50]. More studies that investigate the transcriptome or resistome of bacteria challenged with disinfectants are needed. From a Hazard Analysis and Critical Control Points (HACCP) perspective, utilizing RNA-sequencing could be used to determine critical food safety parameters in a food system environment with the ultimate goal of identifying conditions in food production that mitigate transcription of genes associated with AMR and virulence. From a practical perspective, integrating Whole Genome Sequencing (WGS) and RNA-seq of selected isolates collected during routine surveillance in the processing facility could be used as a way to validate and optimize disinfectant selection. 

Unlike HDP, SHY and TSP, PAA does not have a proposed mechanism for conferring antibiotic resistance or even tolerance (Table 1). Because PAA has two distinct modes of action due to being an organic acid and an oxidant, it is theorized that a cell is less likely to develop tolerance or resistance mechanisms against PAA or antibiotics [40]. This provides valuable information in that in addition to being effective for the control of both *Salmonella* and *Campylobacter* [51], PAA also seems like it is less likely to induce AMR and may even decrease it. One approach could be to utilize PAA at the last step of cleaning and disinfection with the goal of reducing incidence of AMR.

## 5. Antimicrobial Resistance Profiles of Foodborne Pathogens Challenged with Disinfectants

Table 2 provides AMR profiles for HDP, PAA, SHY and TSP, which demonstrates that biocides can differ in the way they induce AMR across different organisms. PAA-challenged *E. coli* resulted in an overall decrease in antimicrobial resistance gene classes [52]. This is in line with the theory that PAA is less likely to induce AMR. HDP-challenged *Salmonella* strains showed a decrease, an increase or both in resistance to certain antibiotics after repeated exposure relative to the wildtype, resulting in mixed effects (Table 2). This emphasizes the importance of testing disinfectants with different serovars and not just single strains of a species as described for *Campylobacter* [30]. One study by Molina-Gonzalez et al. suggests that SHY and TSP can induce AMR when *Salmonella* Enteritidis, *Salmonella* Kentucky and *Salmonella* Typhimurium are exposed to those disinfectants at sub-inhibitory concentrations [36]. Although the experiments from Table 2 cannot be compared directly due to differences in experimental design and analysis, they all provide evidence in support of the conclusion that that proper utilization of disinfectants should include consideration of those biocides that are less likely to increase AMR and biocide tolerance. Therefore, utilizing a serotype-specific approach when selecting disinfectants should be considered by poultry processing facilities. 

## 6. Biofilm Detection

Generally, biofilm-producing strains have been identified quantitatively by microtiter-plate assays or qualitatively by the Congo red agar or test tube methods, both of which use a phenotypic approach [53,54]. Genotypic identification of biofilm-producing strains relies on molecular methods to detect biofilm-associated genes by conventional PCR, qPCR or multiplex PCR [10,55]. The *csgD* gene in *Salmonella* Typhimurium has been identified as a central biofilm regulator gene in which bistable expression allows for either increased virulence or persistence in the environment [50]. Additionally, genes associated with curli, fimbriae, cellulose such as *csgD*, *csgB*, *adrA*, and *bapA* have been utilized to detect *Salmonella* biofilms on eggshells. Furthermore, genes related to flagella adhesion, metabolism, regulation/stress response and proteic envelop/secretion can be used to classify biofilm formation capacity and flagellar motility [18]. 

By using a broad set of phenotypic and genotypic techniques such as the ones mentioned above, it is now well understood that *Salmonella* biofilms are associated with persistence both inside and outside the host including on poultry carcasses and processing plants even after cleaning and disinfection [7,56]. Sensory inspections of open surfaces such as visual, tactile and olfactory observations such as greasy surfaces allow for quick identification of obvious issues in the sanitation process. However, it is important to note that bacterial counts are not correlated with visual inspections [57]. Additionally, while food contact surfaces have become well-established sources of contamination and recontamination in food processing settings [58,59], Arnold and Silvers [60] found that microbial attachment and biofilm formation vary depending on surface type (e.g., stainless steel, conveyor belting, polyethylene and picker-finger rubber). Interestingly, contrary to previous studies that examined planktonic bacteria, picker-finger rubber commonly used in defeathering machines were shown to inhibit microbial contamination and biofilm formation [60]. However, more testing on different combinations of strains observed at the processing plant need to be conducted since it has been shown that microbial attachment and biofilm properties may behave differently depending on the combination of strains that make up the bacterial community [61]. Therefore, based on these considerations, careful and robust assessments of open surfaces at the processing plant should be considered even when it has been well established that *Salmonella* and *Campylobacter* have been isolated from poultry production and processing [9,57]. 

In summary, poultry processing plants should consider taking measures to detect and characterize biofilms from their specific facility to optimize the prevention and management of biofilms. Fortunately, there are now direct and indirect approaches that can be applied at the food processing plant to detect the presence of biofilms through direct observation on open surfaces and to quantify cells isolated from biofilms found at the food processing plant. Direct methods directly observe biofilm colonization, whereas indirect methods start with detaching biofilm from food-contact surfaces before quantifying them [57]. Commercially available and easy-to-use tools that detect the presence of biofilms include BioFinder (Barcelona, Spain) [62], REALCO Biofilm Detection Kit (Louvain-la-Neuve, Belgium) [63], TBF^®^ 300 (Valencia, Spain) and TBF^®^ 300S [64]. TEMPO^®^ system (Marcy l’Etoile, France) allows for quantification via the most probable number (MPN) technique and is also commercially available [65]. Table 3 summarizes biofilm detection methods used in food processing settings. These tools could help improve the eradication of biofilms and can be used to evaluate current cleaning procedures [57]. At the same time, lack of consensus across detection methods should be taken into consideration and a combination of methods for detection and enumeration may need to be implemented [66].

## 7. Biofilm Characterization

In addition to taking measures to detect biofilm-producing bacteria, another important step for processing plants to consider is the characterization of biofilms present at the processing plant. Biofilm characterization can help improve food safety as several differences in biocide susceptibility, pathogenicity and persistence in the environment between biofilms and planktonic bacteria have been identified. 

Minimum inhibitory concentration (MIC) and minimum biofilm eliminating concentrations (MBEC) assays have traditionally been used to determine the efficacy of antibiotics on planktonic and biofilm bacteria, respectively [73]. However, these assays can also be used to determine the efficacy of other biocides such as disinfectants [33]. Furthermore, results from these assays can be used to directly compare planktonic and biofilm bacterial forms. As an example, Chylkova, Cadena, Ferreiro and Pitesky [33] found that acidified calcium hypochlorite (aCH) and PAA were ineffective against *Salmonella* biofilms at contact and concentrations commonly used during poultry processing whereas HDP remained effective, based on MIC and MBEC assays. Similarly, PAA has been found to be inefficient at eliminating *Salmonella* biofilms from polypropylene and polyurethane surfaces which are common surface types used in poultry processing plants [16]. Sarjit and Dykes [74] found that trisodium phosphate (TSP), unlike sodium hypochlorite (SHY), was an effective sanitizer against biofilms on stainless steel, glass and polyurethane surfaces. In contrast, Korber et al. [75] observed *Salmonella* biofilm cells from glass surfaces were less susceptible to TSP. Differences in biofilm response to disinfectants and surfaces indicate further testing is necessary to further elucidate biofilm formation at the processing plant. 

In addition to phenotypic differences as shown by differences in biocide susceptibility, genotypic differences between planktonic and biofilm bacterial cells have also been shown. Wang et al. [76] found that planktonic and biofilm Salmonella Typhimurium cells isolated from raw chicken meat and contact surfaces from poultry processing plants showed distinct gene expression patterns. Specifically, genes from gene ontology groups related to membrane proteins, cytoplasmic proteins, curli productions, transcriptional regulators, cellulose biosynthesis and stress response proteins were differentially expressed suggesting they may play a role in biofilm maturation. Furthermore, virulence and persistence genes have been shown to be differentially expressed in planktonic and biofilm *Salmonella* Typhimurium cells with planktonic cells expressing genes associated with virulence and biofilms expressing genes associated with environmental persistence [50]. These results are in line with results from Borges et al. [77] in which *Salmonella* Typhimurium biofilm production was not associated with in vivo pathogenicity index (PI). However, there was an association between biofilm formation and PI in *Salmonella* Enteritidis at 28 °C [77]. Interestingly, in microaerobiosis and anaerobiosis conditions, *Salmonella* Typhimurium grown in chicken residue displayed downregulation of biofilm associated genes (e.g., *csgD* and *adrA*) and upregulation of virulence genes (e.g., *hilA* and *invA*) on stainless steel [78]. Contrastingly, biofilm formation was upregulated in aerobiosis [78]. Results showed that oxygen levels could have an effect on biofilm formation. Similarly, a study by Wang et al. [79] showed that growth media could also have an effect on biofilm gene expression with biofilm grown in laboratory trypticase soy broth (TSB) expressing upregulation of biofilm formation genes and biofilm grown on meat thawing loss broth (MTLB) expressing inhibited gene expression. 

It is well known that there is a correlation between *Salmonella* biofilm formation and persistence in factory environments; however, several studies have shown that different serovars of *Samonella* have different capacities to make biofilm under different environmental conditions [80,81,82]. For example, when studying *Salmonella* Enteritidis, Infantis, Kentucky and Telaviv serotypes at different temperatures, a shift in biofilm formation capacities was observed with most of the serotypes becoming strong biofilm producers at 22 °C [82]. In contrast, at 37 °C, only some of the *Salmonella* Enteritidis and Infantis serovars were considered strong biofilm producers. In addition to temperature, pH and NaCl concentrations have been shown affect *Salmonella* strains ability to form biofilm [80]. Conditions that were unfavorable and increased biofilm formation in most of the *Salmonella* strains were pH 5.5, 0.5% NaCl and 25 °C [80]. Surface materials also lead to differences in microbial adhesion with polyurethane displaying more irregular adhesion than polypropylene [16]. Moreover, no viable cells were isolated from polypropyle after treatment with sanitizers commonly used in Brazilian poultry processing plants. Overall, these results show that environmental conditions, growth media and strains can influence Salmonella biofilm formation thus highlighting the importance of mimicking food processing conditions during biofilm and disinfectant efficacy testing. 

In summary, the persistence and complexity of *Salmonella* biofilms in food processing plants suggest that protocols used to control and eliminate biofilms should be constantly evaluated and modified accordingly. This is especially relevant when considering that many of the cleaning products used in the food industry are not optimized for the elimination of biofilms [83] and an estimated 65% of human bacterial infections are caused by biofilms [84]. It is also important to note that in vitro efficacy testing of disinfectants should be done on mature biofilms as that is the bacterial form and stage most commonly found in food processing environments particularly food-contact surfaces [85]. Likewise, testing conditions should mimic poultry processing conditions in order to improve the applicability of the results as strain, environmental conditions and growth mediums have been shown to influence biofilm formation. Furthermore, alternative biofilm eradication methods that act specifically on biofilms should also be considered, such as lactic acid bacteria, phagetherapy, crude essential oils, quorum sensing inhibitors and bacteriocins [10,86]. However, it should also be noted that the most effective strategy would be to prevent biofilm formation in the first place [10]. 

## 8. Conclusions

While more research is needed to further our understanding of AMR profiles from pathogens isolated from poultry processing facilities, this review suggests that understanding what AMR mechanisms are activated by disinfectants can provide poultry processing facilities insights as to which disinfectants to use at a particular facility. Therefore, active monitoring of pathogens present at the grow-out facility and utilizing that information to strategize which disinfectants to employ at the processing plant (i.e., serotype-specific approach) should be considered. The efficacy of disinfectants on biofilms in addition to planktonic bacteria should be frequently tested in order to monitor for changes in susceptibility to disinfectants and prevent the use of sub-inhibitory concentrations. Using disinfectants that differ in their modes of action throughout poultry processing is also advisable as this can potentially reduce the ability of the bacteria to adapt and become tolerant to biocides and antimicrobials. The effect of the disinfectants on the transcriptome or resistome of the pathogen in question may be the key to furthering our understanding of AMR. Alternative approaches to the control of planktonic bacteria and biofilm formation that do not rely on the use of traditional biocides, such as enzymes, bacteriophages and quorum sensing inhibitors may be valuable to controlling microbial contamination without inducing AMR, and thus may be the new horizon of antimicrobial food safety [7,9,10,13,87].

## Figures and Tables

**Figure 1 foods-08-00275-f001:**
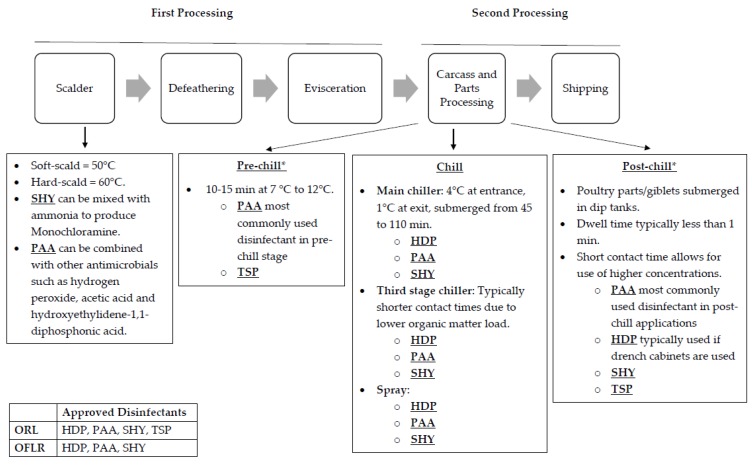
A general schematic overview of commercial poultry processing. The information provided serves as an example of possible scenarios. However, protocols can vary widely among processing plants. In addition, the list of antimicrobials approved for on-line reprocessing (OLR) and off-line reprocessing (OFLR) is dynamic in terms of application and concentration. * At a minimum, sampling for pathogens must occur at the pre- and post-chill points.

**Table 1 foods-08-00275-t001:** Mode of action (MOA) and proposed mechanism of antibiotic resistance for the following commonly used disinfectants in poultry processing: HDP, PAA, SHY and TSP (HDP: hexadecylpyridinium chloride, PAA: paracetic acid, SHY: sodium hypochlorite, TSP: trisodium phosphate).

Disinfectant	Disinfectant Type	Proposed Modes of Action	References	Proposed Mechanism Conferring Antibiotic Resistance	References
HDP	Quaternary ammonium	Adsorption and penetration of cell wall. Disruption of cytoplasmic membrane. Leakage of intracellular low molecular-weight constituents. Degradation of proteins and nucleic acids. Cell lysis due to cell wall degrading autolytic enzymes.	[30,39]	Overexpression of efflux pumps. Induce cellular morphological changes such as thickening of cell envelope or loss in outer membrane proteins.	[39]
PAA	Organic acid and an oxidant	Non-specific oxidation particularly of C–C double bonds and reduced atoms (i.e., S).	[40]	None known.	[40]
SHY	Chlorine	Uncoupling of the electron chain or enzyme inactivation (i.e., trans-phosphorylase inactivation) either in the membrane or in the cell interior.	[41]	Induces biofilm formation.	[42]
TSP	Alkaline detergent	High pH (12 to 13) disrupts cytoplasmic and outer membranes resulting in leakage and eventual cell death. High ionic strength can cause bacterial cell autolysis. Removes bacterial cells from carcass surface (i.e., chicken skin) by removing a thin layer of lipids (“detergent” effect) from the surface of the carcass thereby exposing cells that would otherwise be protected, and results in bacterial cell autolysis.	[30,41]	Overexpression of efflux pumps. Induce cellular morphological changes such as loss in outer membrane proteins.	[30]

HDP: hexadecylpyridinium chloride, PAA: peracetic acid, SHY: sodium hypochlorite, TSP: trisodium phosphate.

**Table 2 foods-08-00275-t002:** AMR profiles of isolates challenged with disinfectants commonly used during poultry processing.

Reference	Disinfectant	Application Parameters	Isolate	AMR Profile			
[29]				**Showed increased resistance (i.e., twice the MIC) to the following antimicrobials compared to the wildtype:**		**Showed decreased resistance to the following antimicrobials compared to the wildtype:**	
	HDP	Exposed to increasing concentrations of CPC (0.01, 0.1, 1, 5, 10, 50, 100, 200, 500 mg/mL, 1, 2, 5 and 10 mg/mL).	*Salmonella* UJA59l	Ampicillin, Sulfamethoxazole, Nalidixic acid		Ceftazidime	
			*Salmonella* UJA82k			Ceftazidime	
			*Salmonella* UJA82l	Nalidixic acid		Ampicillin, Cefotaxime, Ceftazidime, Sulfametoxazol	
[4]				**HDP tolerance level:**		**Showed resistance to:**	
	HDP	HDP tolerance and antibiotic resistance were determined by using MIC assays.	*Salmonella* spp. UJAS6	Tolerant		Ampicillin, Chloramphenicol, Tetracycline, Nalidixic acid, Trimethoprim-sulfamethoxazole	
			*S. enterica* UJAS10	Tolerant		Ampicillin, Tetracycline, Nalidixic acid, Trimethoprim-sulfamethoxazole	
			*Salmonella* spp. UJAS18	Tolerant		Ampicillin, Cefotaxime; Ceftazidime, Ciprofloxacin, Chloramphenicol, Tetracycline, Netilmicin, Nalidixic acid, Trimethoprim-sulfamethoxazole	
			*Salmonella* spp. UJAS19	Tolerant		Cefotaxime; Ceftazidime, Ciprofloxacin, Chloramphenicol, Streptomycin, Tetracycline, Netilmicin, Nalidixic acid, Trimethoprim-sulfamethoxazole	
[30]	HDP	Step-wise exposure to gradually increasing concentrations (2, 2.5, 3, 4 to 5 mg/mL, depending upon the growth of the adapted microorganism) of HDP over 15 days.		**MIC fold change of *Campylobacter* strains relative to the pre-adapted strains.**			
			*Campylobacter jejuni* K49/4	Days after repeated exposure to HDP	5	10	15
				MIC fold change	1	1	1
			*Campylobacter jejuni* NCTC11168	Days after repeated exposure to HDP	5	10	15
				MIC fold change	2	1	4
			*Campylobacter jejuni* ATCC33560	Days after repeated exposure to HDP	5	10	15
				MIC fold change	0.5	1	1
			*Campylobacter coli* 137	Days after repeated exposure to HDP:	5	10	15
				MIC fold change	1	1	0.5
			*Campylobacter coli* ATCC33559	Days after repeated exposure to HDP	5	10	15
				MIC fold change	1	2	2
[52]				**Number of antimicrobial resistance gene classes in PAA treated strains:**			
	PAA	Exposed to 0.9 to 2.0 mg/L of PAA to reach target disinfection level of 200 CFU/100mL	*Escherichia coli*	Mean number of classes decreased by an average of 47% with significant reductions in the following classes: Macrolides (−62.3%), Beta-lactams (−41.3), Phenicols (−64) and Trimethoprim (−49.9).			
[36]				**Showed increased resistance (i.e., susceptible to resistant via disk diffusion assay) after exposure to disinfectant:**			
	SHY	Exposed to increasing sub-inhibitory concentrations (starting at MIC/2).	*Salmonella* Enteritidis	Ceftazidime			
			*Salmonella* Kentucky	Amikacin, Ampicillin/ sulbactam			
			*Salmonella* Typhimurium	Amikacin, Tobramycin, Cefazolin, Cefotaxime			
	TSP		*Salmonella* Enteritidis	Amikacin, Cefazolin, Cefoxitin, Ceftazidime, Aztreonam, Nalidixic acid, Phosphomycin			
			*Salmonella* Kentucky	Amikacin, Ceftazidime, Aztreonam, Phosphomycin			
			*Salmonella* Typhimurium	Amikacin, Cephalothin, Cefazolin, Cefoxitin, Cefepime, Aztreonam, Phosphomycin			
[31]				**Mean number of antibiotics the strains were resistant to at 0 days of storage:**		**Mean number of antibiotics the strains were resistant to after 5 days of storage:**	
	TSP	Chicken legs containing *E. coli* were dipped in 12% TSP at 20 ± 1 °C for 15 min and subsequently refrigerated at 7 ± 1 °C and stored. Chicken legs dipped in tap water were used as a control.	*Escherichia coli*	Control: 3.76 ± 2.01 ^a^_a_ TSP: 3.80 ±2.48 ^a^_a_		Control: 3.44 ± 1.42 ^a^_a_ TSP: 4.64 ± 2.64 ^b^_b_	
				The mean numbers from the same day (different treatments) with no letters in common (superscript) are significantly different (*P* < 0.05). The mean numbers within the same treatment (day 0 versus day 5) with no letters in common (subscript) are significantly different (*P* < 0.05).			
[30]				**MIC fold change of *Campylobacter* strains relative to the pre-adapted strains.**			
	TSP	Step-wise exposure to gradually increasing concentrations (2, 2.5, 3, 4 to 5 mg/mL, depending upon thegrowth of the adapted microorganism) of TSP over 15 days.	*Campylobacter jejuni* K49/4	Days after repeated exposure to TSP	5	10	15
				MIC fold change	2	2	2
			*Campylobacter jejuni* NCTC11168	Days after repeated exposure to TSP	5	10	15
				MIC fold change	2	0.5	2
			*Campylobacter jejuni* ATCC33560	Days after repeated exposure to TSP	5	10	15
				MIC fold change	1	1	0.125
			*Campylobacter coli* 137	Days after repeated exposure to TSP	5	10	15
				MIC fold change	1	0.008	0.004
			*Campylobacter coli* ATCC33559	Days after repeated exposure to TSP	5	10	15
				MIC fold change	2	2	1

HDP: hexadecyzpyridinium chloride; PAA: peracetic acid; SHY: sodium hypochlorite; TSP: trisodium phosphate.

**Table 3 foods-08-00275-t003:** Direct and indirect biofilm detection and enumeration methods for food processing settings.

Test	Type	Method	References
**Direct**			
BioFinder	Qualitative	Direct observation of color change due to dying of biofilm components.	[62]
Contact plates	Quantitative	Sterile agar plate is placed on surface of interest and biofilm is detected via conventional culture methods.	[67]
Direct epifluorescence microscopy	Quantitative	Automatic cell quantification using computer software on digital images.	[68]
REALCO Biofilm Detection Kit	Qualitative	Direct observation of color change due to dying of biofilm components.	[63]
TBF^®^ 300/ TBF^®^ 300S	Qualitative	Direct observation of color change due to dying of biofilm components.	[64]
**Indirect**			
BacTrac 4300	Quantitative	Total viable counts calculated via impedance.	[69,70]
Plate count	Quantitative	Culture plating to determine the number of colony forming units (CFU).	[57]
TEMPO^®^	Quantitative	Cell counts from biofilms are calculated using most probable number (MPN) system based on fluorescence.	[65]
Abcam XTT tetrazolium salt and resazurin assay kit	Quantitative	Metabolic assays combined with spectrophotometry can be used to quantify biofilm.	[57,71,72]

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
