# Peer review of "Understanding Antimicrobial Resistance (AMR) Profiles of Salmonella Biofilm and Planktonic Bacteria Challenged with Disinfectants Commonly Used During Poultry Processing"

_foods, 2019, doi:10.3390/foods8070275_

Reviewer 1 Report

The revision made to the paper addressed the small concerns I had in the first revision. The authors made the changes necessary and clarified the aspects that I had doubts about. Thus, I consider that the manuscript able to be accepted.

Author Response

Not applicable.

Reviewer 2 Report

According to the revised manuscript, some of my previous comments were not addressed.

line 56: please check punctuation and spacing
Line 79, 80, 83: "Two methods of reprocessing include (OLR) and (OFLR)". - why abbreviation is in brackets?
if these abbreviated names were used why in the next two sentences the full name is given? - it does not make sense.

Author Response

REVIEWER #2:

1. Line 56: please check punctuation and spacing.

Original:

Additionally, biofilm formation confers increased antimicrobial resistance through a variety of mechanisms,. From a practical perspective,biofilm cells are 10 to 1000 fold less susceptible to anti-microbial agents than the planktonic form of the same bacterium [8,14,15]. 

Response to reviewer:

Thank you for your help. Punctuation and spacing are now fixed.

Updated:

Additionally, biofilm formation confers increased antimicrobial resistance through a variety of mechanisms. From a practical perspective, biofilm cells are 10 to 1000 fold less susceptible to anti-microbial agents than the planktonic form of the same bacterium [8,14,15].

2. Line 79, 80, 83: "Two methods of reprocessing include (OLR) and (OFLR)". - why abbreviation is in brackets? if these abbreviated names were used why in the next two sentences the full name is given? - it does not make sense.

Original:

Two methods of reprocessing include (OLR) and (OFLR). Processing plants may use either or both forms of reprocessing. In on-line reprocessing, contaminated carcasses can be reprocessed manually by trimming away contaminated parts and/or treated with an antimicrobial along with poultry carcasses that are not contaminated on the main line. In contrast, off-line reprocessing entails taking out contaminated carcasses off the main line where they can be reprocessed manually by trimming away contaminated parts and/or treated with an antimicrobial, away from the visually uncontaminated carcasses. Then, before carcass chilling, carcasses are visually inspected again for fecal contamination using guidelines provided by the Food Safety Inspection and Inspection Service Directive in order to comply with the zero tolerance standard that requires carcasses to free of fecal contamination before entering the chiller tank [22,23]. While disinfectants are used in both first and second processing, their application, contact time and temperature may differ. Similarly, disinfectants approved for use during on-line and off-line reprocessing may differ in the application, contact time and temperature [24]. For the purposes of this review, only disinfectants commonly used during poultry processing will be reviewed as intervention protocols may vary greatly across poultry processing plants. Popular disinfectants include hexadecylpyridinium chloride (HDP), peracetic acid (PAA), sodium hypochlorite (SHY) and trisodium phosphate (TSP) [25,26]. While PAA is the most commonly used disinfectant in both pre- and post-chill applications, HDP is most commonly used in post-chill applications when drench cabinets are used [27]. Additionally, both HDP and PAA can be used for on-line OLR and OFLR  [24]. SHY and TSP are also commonly used in post-chill applications [25]. In terms of reprocessing, SHY is only allowed in OFLR, although when it is used in combination with other disinfectants it can be used in OLR as well. In contrast, TSP is only allowed in OLR. Figure 1 provides a schematic overview of poultry processing along with information on HDP, PAA, SHY and TSP.

Response to reviewer:

Thank you for catching these inconsistencies. Acronyms have been fixed

Updated:

Two methods of reprocessing include on-line reprocessing (OLR) and off-line reprocessing (OFLR). Processing plants may use either or both forms of reprocessing. In OLR, contaminated carcasses can be reprocessed manually by trimming away contaminated parts and/or treated with an antimicrobial along with poultry carcasses that are not contaminated on the main line. In contrast, OFLR entails taking out contaminated carcasses off the main line where they can be reprocessed manually by trimming away contaminated parts and/or treated with an antimicrobial, away from the visually uncontaminated carcasses. Processing plants may use either or both forms of reprocessing. Then, before carcass chilling, carcasses are visually inspected again for fecal contamination using guidelines provided by the Food Safety Inspection and Inspection Service Directive in order to comply with the zero tolerance standard that requires carcasses to free of fecal contamination before entering the chiller tank [22,23]. While disinfectants are used in both first and second processing, their application, contact time and temperature may differ. Similarly, disinfectants approved for use during OLR and OFLR may differ in the application, contact time and temperature [24]. For the purposes of this review, only disinfectants commonly used during poultry processing will be reviewed as intervention protocols may vary greatly across poultry processing plants. Popular disinfectants include hexadecylpyridinium chloride (HDP), peracetic acid (PAA), sodium hypochlorite (SHY) and trisodium phosphate (TSP) [25,26]. While PAA is the most commonly used disinfectant in both pre- and post-chill applications, HDP is most commonly used in post-chill applications when drench cabinets are used [27]. Additionally, both HDP and PAA can be used for OLR and OFLR  [24]. SHY and TSP are also commonly used in post-chill applications [25]. In terms of reprocessing, SHY is only allowed in OFLR, although when it is used in combination with other disinfectants it can be used in OLR as well. In contrast, TSP is only allowed in OLR. Figure 1 provides a schematic overview of poultry processing along with information on HDP, PAA, SHY and TSP.

This manuscript is a resubmission of an earlier submission. The following is a list of the peer review reports and author responses from that submission.

Round  1

Reviewer 1 Report

The article #451878 entitled “Understanding Antimicrobial Resistance (AMR) 2 Profiles of Salmonella Biofilms Challenged with 3 Disinfectants Commonly Used During Poultry 4 Processing” is a very interesting review, related to a subject of maximum concern nowadays.

Authors reviewed, from an adequate and actual number of research papers and conceptual literature the problem of resistance to antimicrobials in salmonella. The problem of cross resistance between biocides and antibiotics is clearly focused, stressing out the importance of have a serious concern on sub-lethal use of disinfectants in poultry industry.

As a non-native English speaker, I found the language very correct to an international journal.

I just have a few small notes that could improve the article.

Notes: In the legend of Figure 1, It could be useful to use the full name for OLR and FOLR, once that full name only appears in the text after the figure.

Line 76 – pas inspection. I believe that pas is missing one “s”

Line 104 – can carry – I suggest “have”, once besides the contamination with organic matter, the carcass is organic matter too.

Line 137-138 and 167-168 – I believe that the international reader of this artcle knowns that campylobacter and Escheria are Gram-negatives. I am not sure if the note of the authors is really necessary.

Line 179-182. I understood the meaning of this relationship authors want to establish between RNA sequencing and HACCP. I my opinion a little more speculation on how these laboratory approaches might be useful in HACCP could improve the interest of calling this relationship to the review. Should we consider the presence of resistance as a hazard besides the presence of salmonella itself? What kind of preventive measures might be applied, in conformity with the HACCP method. My concern is the following: How will a technician of the poultry industry responsible for HACCP system look to this information?

Line 213. Additionally could de deleted, and avoids the repletion wuith the addition in the phrase. . Line 219 and 221 – “may be the key”. I sugest to merge both phrases to avoid the repetition

Author Response

Line 76 – pas inspection. I believe that pas is missing one “s”

Response: Correction made

Line 104 – can carry – I suggest “have”, once besides the contamination with organic matter, the carcass is organic matter too.

Response: Correction made

Line 137-138 and 167-168 – I believe that the international reader of this artcle knowns that campylobacter and Escheria are Gram-negatives. I am not sure if the note of the authors is really necessary.

Response:

The focus of this manuscript is on Salmonella but we also discuss Campylobacter and Pseudomonas which are also Gram-negative. Although most readers know this we wanted to make this association clear in the manuscript.

Response:

Line 179-182. I understood the meaning of this relationship authors want to establish between RNA sequencing and HACCP. I my opinion a little more speculation on how these laboratory approaches might be useful in HACCP could improve the interest of calling this relationship to the review. Should we consider the presence of resistance as a hazard besides the presence of salmonella itself? What kind of preventive measures might be applied, in conformity with the HACCP method. My concern is the following: How will a technician of the poultry industry responsible for HACCP system look to this information?

Response: The following additional sentence was added starting at line 182: From a practical perspective, integrating Whole Genome Sequencing (WGS) and  RNA-seq  of selected isolates collected during routine surveillance in the processing facility could be used as a way to validate and optimize disinfectant selection.

Line 213. Additionally could de deleted, and avoids the repletion wuith the addition in the phrase. . Line 219 and 221 – “may be the key”. I sugest to merge both phrases to avoid the repetition

Response: Additionally was deleted. Due to the length of the individual sentences the 2 sentences were not merged.

Reviewer 2 Report

Title: Understanding Antimicrobial Resistance (AMR) Profiles of Salmonella Biofilms Challenged with Disinfectants Commonly Used During Poultry Processing

The manuscript reviews how commonly used industrial disinfectants influence bacterial susceptibility throughout poultry processing and mechanisms of AMR development are described. In general, the manuscript is well structured and the topic indeed can contribute to the field, however, the language, grammar should be improved significantly. Below are suggestions, which can be used as a guide for further improvements.

Line 22: change “antimicrobial resistance” to AMR as the abbreviation has been provided earlier in line 15.

Line 43: bacterial colonies – change to bacterial cells

Line 46: Full stop after “bacterial growth”

Line 48: change to “which are regulated…”

Line 52-54: the authors should revise this sentence and check the corresponding references.  “to encasing bacteria” should be removed; “increases nutrient availability, metabolic enhancement” seems misleading.

Line 55: “Additionally, biofilm formation confers increased antimicrobial resistance through a variety of mechanisms, such that” should be removed as the next statement is not relevant to the mechanisms.

Line 57: the use

Line 59 and elsewhere: Salmonella - should be italicised

Line 59 – 64: this sentence is too long. A definition of quorum sensing can be a separate sentence, as well as the “development of QS inhibitors…”

Line 70: Figure 1: Please indicate at which stage/s microbiological sampling occur.

Line 73: In the legend for figure 1, please use full name for OLR and OFLR

Line 77: “In on-line reprocessing” - use abbreviation

Line 79: “off-line reprocessing“ – use abbreviation

Line 77 - 81: regarding OLR and OFLR – please specify on the basis of which criteria the type of reprocessing is selected. Please schematically depict sampling points for micro assessment in Figure 1.

Line 81: Please provide more details on how visual assessment or what criteria for it are used to determine/identify whether the carcases are contaminated or uncontaminated.

Line 99: change resistance to resistant

Line 101 – 104: this sentence should be revised/rewritten as it does not make sense.

Line 108: “matrix limits access” please specify to what, biocides? The whole sentence should be revised as it does not sound right

Line 111: easier to kill – substitute it to more scientifically sound word.

Line 111-112: move down (see comment below) “While biofilms are more prevalent than the easier to-kill planktonic form in processing plants,”

Line 113: Delete “and or log reduction of the” - inhibitory concentration of biocide against bacteria in their planktonic form.

Line 114 – 115: Delete “if the tests used to determine effective disinfectant 114 concentrations are done on planktonic bacteria,” it is redundant

Line 115: “Since biofilms are more prevalent than the easier to-kill planktonic form in processing plants, it is possible that the concentrations are sub inhibitory in the processing plant.”

Line 147: please provide an examples of these biocides

Line 202: delete one “that”

Author Response

Line 22: change “antimicrobial resistance” to AMR as the abbreviation has been provided earlier in line 15.

Response: Correction made

Line 43: bacterial colonies – change to bacterial cells

Response: Correction made

Line 46: Full stop after “bacterial growth”

Response: Correction made

Line 48: change to “which are regulated…”

Response: Correction made

Line 52-54: the authors should revise this sentence and check the corresponding references.  “to encasing bacteria” should be removed; “increases nutrient availability, metabolic enhancement” seems misleading.

Response: encasing bacteria was rephrased as “to the enclosed bacteria.”

Line 55: “Additionally, biofilm formation confers increased antimicrobial resistance through a variety of mechanisms, such that” should be removed as the next statement is not relevant to the mechanisms.

Response: Correction made. The sentence was split into two sentences

Line 57: the use

Response: Correction made

Line 59 and elsewhere: Salmonella - should be italicized

Response: Correction made

Line 70: Figure 1: Please indicate at which stage/s microbiological sampling occur.

Response: A sentence was added at line 74 indicating that sampling for pathogens must occur at a minimum at the pre- and post-chill points. “*At a minimum, sampling for pathogens must occur at the pre- and post-chill points.” Asterisks were added next to the pre- and post-chill headings in the diagram.

Line 73: In the legend for figure 1, please use full name for OLR and OFLR

Response: Correction made

Line 77: “In on-line reprocessing” - use abbreviation

Response: Correction made

Line 79: “off-line reprocessing“ – use abbreviation

Response: Correction made

Line 77 - 81: regarding OLR and OFLR – please specify on the basis of which criteria the type of reprocessing is selected. Please schematically depict sampling points for micro assessment in Figure 1.

Response: This is not a FSIS or similar regulatory decision but more of a decision based upon what reprocessing method the company may prefer. We added the following line at line 82 “Processing plants may use either or both forms of reprocessing.”

Line 81: Please provide more details on how visual assessment or what criteria for it are used to determine/identify whether the carcases are contaminated or uncontaminated.

Response: In line 75 we say “contaminated with fecal matter or digestive tract content.” However, we modified the sentence in line 75 to give more information on when the inspection occurs. In line 75, “Moreover, during processing, carcasses that are contaminated with fecal matter or digestive tract content as determined by visual inspection right after evisceration may still pass inspection if reprocessed properly [21].

Added the following starting at line 88, “Then, before carcass chilling, carcasses are visually inspected again for fecal contamination using guidelines provided by the Food Safety Inspection and Inspection Service Directive in order to comply with the zero tolerance standard that requires carcasses to free of fecal contamination before entering the chiller tank [22.23].

Line 99: change resistance to resistant

Response: Correction made

Line 101 – 104: this sentence should be revised/rewritten as it does not make sense.

Response: the sentence was revised to the following:

In addition, challenging Salmonella with sub-inhibitory concentrations of disinfectants at the processing plant can occur due to the presence of high loads of organic material which can inactivate the disinfectant, inadequate distribution or concentration of the disinfectant and  high concentration of biofilm(s), ,.

Line 108: “matrix limits access” please specify to what, biocides? The whole sentence should be revised as it does not sound right

Response: We are giving a general description of the matrix and how it limits access to “stressors” like antibiotics and disinfectants.  

Line 111: easier to kill – substitute it to more scientifically sound word. The sentence was revised to the following “With respect to biofilms, the extracellular matrix limits access to stressors, thereby providing some protection against disinfectants and antibiotics that allow Salmonella to  persist on biotic (e.g. live birds) and abiotic (e.g. stainless steel) surfaces [6].

Response: sentence changed to “While biofilms are more prevalent than there planktonic form in processing plants, the common methodology for evaluating the efficacy of disinfectants requires identifying the inhibitory concentration and or log reduction of the bacteria in their planktonic form [31].”

Line 111-112: move down (see comment below) “While biofilms are more prevalent than the easier to-kill planktonic form in processing plants,”

Response: Correction made based on comments that were made three comments down “Line 115: “Since biofilms…”

Line 113: Delete “and or log reduction of the” - inhibitory concentration of biocide against bacteria in their planktonic form.

Response: Correction made

Line 114 – 115: Delete “if the tests used to determine effective disinfectant 114 concentrations are done on planktonic bacteria,” it is redundant

Response: Correction made

Line 115: “Since biofilms are more prevalent than the easier to-kill planktonic form in processing plants, it is possible that the concentrations are sub inhibitory in the processing plant.”

Response: Correction made

Line 147: please provide an examples of these biocides

Response: We included the biocides tested by Mavri et al 2013.(starting at line 158) “Along with linking efflux pumps to cross-resistance, this study revealed that mechanisms involved in biocide (e.g. triclosan, benzalkoniumchloride, hexadecylpyridinium chloride, chlorhexidine diacetate and trisodium phosphate) adaptation are unique for various strains of Campylobacter as opposed to it having only one species-specific mechanism [28].”

Line 202: delete one “that”

Response: Correction made

Reviewer 3 Report

This paper ("Understanding Antimicrobial Resistance (AMR) Profiles of Salmonella Biofilms....") is a review of information regarding the application of biocides and disinfectants in poultry processing with un-intended consequences of affecting antimicrobial resistance. Overall, the paper is well written and contains numerous references supporting the material. The topic is not foreign to my own experiences with antimicrobial interventions and I was glad to see that the authors eventually brought in the notion that sub-lethal levels of antimicrobials that can promote proliferation of AMR Salmonella can also be caused simply by the organic substrate they are applied on (poultry surfaces, soluble protein in solution in dip tanks, etc.) that quickly eliminates oxidants (hypochlorite) or neutralize acidic antimicrobials. I appreciate the recommendations the authors  make with suggestions that "serotype-/strain-specific approaches to disinfectants" (Line 151) or "the use of enzymes, bacteriophage, and/or quorum-sensing inhibitors" (Line 221) are likely to become more important or widely used in the future.

My main issue with the paper is the title (inclusion of the word 'biofilms' in particular) with the content of the paper which mostly is concerned with poultry processing. Although I agree, biofilms present a formidable challenge to penetration of antimicrobials (as well as nutrients, etc), I do not agree that the main reason poultry carcasses fail to get decontaminated is because of biofilms on biotic surfaces (I believe it's more the result of neutralization of the biocide) and there is little that the authors have presented to persuade me of that. I have done many meat/poultry surface inoculations (20-60 min) that are sprayed/dipped with antimicrobials and yet achieve minimal reductions, not because of biofilms, but because of either too low of biocide level or the biocide being inactivated. 

Specific corrections:

Remove 'biofilm' from the title.

Line 76: Misspelling:  "….may still pass inspection...."

Section 3 (Lines 95-125): The authors refer to Salmonella in biofilms being protected from both 'biocides' and 'antibiotics' during poultry processing and it is not clear where 'antibiotics' are being applied in this regard?  Are they referring to antibiotics applied as growth promotants in animal feed?, something which is becoming less and less common as it is being attacked by consumer groups and large end-users with clout (i.e., McDonalds') are advertising they are not using animals grown with such practices. It sounds as if 'antibiotics' (i.e., the clinical antibiotics, not generic reference to antimicrobials) are being used as antimicrobial spray treatment for Salmonella.

Line 137:  "Gram-negative", "Gram" is the name of a person and should be capitalized.

Line 139: Rephrase? ,  "....and by evaluating their outer membrane proteins." A bit awkward ending of the sentence; sounds as if something should continue after this?

Author Response

My main issue with the paper is the title (inclusion of the word 'biofilms' in particular) with the content of the paper which mostly is concerned with poultry processing. Although I agree, biofilms present a formidable challenge to penetration of antimicrobials (as well as nutrients, etc), I do not agree that the main reason poultry carcasses fail to get decontaminated is because of biofilms on biotic surfaces (I believe it's more the result of neutralization of the biocide) and there is little that the authors have presented to persuade me of that. I have done many meat/poultry surface inoculations (20-60 min) that are sprayed/dipped with antimicrobials and yet achieve minimal reductions, not because of biofilms, but because of either too low of biocide level or the biocide being inactivated. 

Response: Very interesting comment. Re: I do not agree that the main reason poultry carcasses fail to get decontaminated is because of biofilms on biotic surfaces (I believe it's more the result of neutralization of the biocide).

From our work using the FSIS-GRAS allowed levels of several disinfectants including PAA, CPC and aCH, we have examples of Salmonella biofilms which were resistant in laboratory settings on abiotic surfaces which presumably should confer an ‘advantage’ to the disinfectant relative to the biofilm. From our previous work, there are concentrations we can eliminate biofilms using some of these disinfectants but the concentrations were well above the regulatory levels.

Specific corrections:

Remove 'biofilm' from the title.

In order to address the comment regarding the title we changed the title to cover planktonic and biofilm. Specifically we changed the title to “ Understanding Antimicrobial Resistance (AMR) Profiles of Salmonella Biofilm and Planktonic Bacteria Challenged with Disinfectants Commonly Used During Poultry Processing” We believe this title appropriately reflects this review.

Line 76: Misspelling:  "….may still pass inspection...."

Response: Correction made

Section 3 (Lines 95-125): The authors refer to Salmonella in biofilms being protected from both 'biocides' and 'antibiotics' during poultry processing and it is not clear where 'antibiotics' are being applied in this regard?  Are they referring to antibiotics applied as growth promotants in animal feed?, something which is becoming less and less common as it is being attacked by consumer groups and large end-users with clout (i.e., McDonalds') are advertising they are not using animals grown with such practices. It sounds as if 'antibiotics' (i.e., the clinical antibiotics, not generic reference to antimicrobials) are being used as antimicrobial spray treatment for Salmonella.

Response: While “growth promoting” doses of antibiotics in grow-out poultry farms are no longer legal in the U.S. and other countries there are still places where the addition of antibiotics at growth promoting doses is very common. In addition non-growth promoting doses are still in use in conventional poultry production as a response to various outbreaks of bacterial disease (primarily Colibacillosis). Therefore we believe the section is appropriately written.

Line 137:  "Gram-negative", "Gram" is the name of a person and should be capitalized.

Response: Correction made

Line 139: Rephrase? ,  "....and by evaluating their outer membrane proteins." A bit awkward ending of the sentence; sounds as if something should continue after this?

Response: Rephrased in line 146 to to “In order to evaluate these proposed mechanisms, Mavri et al. exposed TSP- and HDP- adapted Campylobacter jejuni and Campylobacter coli (Gram-negative bacteria like Salmonella, which share a superfamily of efflux pumps known as Resistance-Nodulation-Division [35,36]) to efflux pump inhibitors and evaluated their outer membrane proteins.”